# Effects of School-Based Exercise Program on Obesity and Physical Fitness of Urban Youth: A Quasi-Experiment

**DOI:** 10.3390/healthcare9030358

**Published:** 2021-03-22

**Authors:** Ji Hwan Song, Ho Hyun Song, Sukwon Kim

**Affiliations:** 1Department of Physical Education, Jeonju National University of Education, Jeonju-si 55101, Korea; ufosong114@jnue.kr; 2Ho-Sung Middle School, Jeollabukdo Office of Education, Jeonju-si 54817, Korea; hohyunss@jbedu.kr; 3Department of Physical Education, Jeonbuk National University, Jeonju-si 54896, Korea

**Keywords:** Physical Activity Promotion System, physical fitness, BMI, adolescents, school health

## Abstract

(1) Purpose: The purpose of this study was to evaluate if an after-school intervention program could prevent obesity and promote the physical fitness of urban sedentary school children. (2) Methods: A 16-week after-school physical fitness program was provided to 36 middle school students (7th, 8th, and 9th graders) recruited from a middle school for three days a week. They were high-risk youth showing poor health status (level 4 or 5) in the regular physical fitness evaluation conducted. Their body mass index (BMI), cardiovascular endurance, muscular strength and endurance, quickness, and flexibility were evaluated. (3) Results: A paired sample *t*-test was used (α = 0.05). There were statistical differences ((x ± s), *p* < 0.05) between the pre-Progressive Aerobic Cardiovascular Endurance Run (PACER) (13.36 ± 4.98 (# of laps)) and post-PACER (18.64 ± 6.31 (# of laps)) (*p* < 0.001), between the pre-sit-up (18.06 ± 7.22 (# of sit-ups)) and post-sit-up (24.89 ± 7.52 (# of sit-ups)) (*p* < 0.001), and between the pre-Trunk Flexion (2.64 ± 3.49 ((cm))) and post-Trunk Flexion (5.97 ± 2.78 ((cm)) (*p* < 0.001). There was no statistical difference between the pre-50m-Run (10.74 ± 1.30 ((sec))) and post-50m-Run results (10.69 ± 1.25 (sec)) (*p* = 0.063) or between the pre-BMI (24.84 ± 3.97 (kg/m^2^)) and post-BMI (24.76 ± 3.61 (kg/m^2^)) (*p* = 0.458). Overall, the physical fitness measures improved, whereas BMI did not change after 16 weeks. (4) Conclusion: Sixteen weeks of the school-based Health-related Physical Fitness (HrPF) program can be effective in improving overall physical fitness levels of adolescents, although additional treatments would be required to change BMI, which showed no improvement in the present study. It is concluded that in order for adolescents to maintain and promote physical fitness and health, participation in a school-based HrPF program is recommended for at least 30 min a day, three days a week. In addition, schools should provide high-risk youth with easy access to physical activities that are similar to the objectives of the physical education curriculum. In order to promote the health of school-age children, each school should establish and operate school-based systematic intervention programs.

## 1. Introduction

Regular participation in physical activity (PA) has numerous physical benefits in adolescents, promoting bone and muscle growth, helping maintain proper weight and body composition, and contributing to disease prevention such as high blood pressure, diabetes, and obesity [1,2,3]. It also has mental health benefits, as it activates various areas of the brain, promotes self-efficacy, and improves resilience against, among other things, stress, anxiety, and depression. [4,5]. Adolescents who maintain an appropriate level of physical activity increase their likelihood of remaining active into adulthood [6,7,8,9]. On the other hand, inactivity may lead to the development of chronic diseases such as high blood pressure and diabetes during adulthood [9]. Adolescents with poor physical health status can benefit more from regular participation in PA [1,8]. To achieve health benefits for high-risk youth or school-age children, it is important to encourage them to be physically active.

As mobile devices such as smartphones, computers, tablets, video games, and social media are incorporated into the lives of youths, the PA level has begun to gradually decrease and will continue to decline [1,6,8]. Schoolchildren and adolescents spend most of their time each day at school, so the role of school should have a great influence on their health [1,6,8]. However, the scope, quantity, quality, and participation rate in physical education courses are currently not at the desired levels in most of schools [1,6,8]. Schools should offer good educational support that results in the positive attitudes necessary for lifelong PA habits [1,6]. To meet the exigency of improving adolescent health and physical fitness, a measurement and evaluation system that emphasizes physical fitness testing and physical activity related education should be provided [8]. The Physical Activity Promotion System (PAPS) was employed in accordance with these needs in Korea [10,11].

The ministry of Education in Korea enforced PAPS at all public and private elementary schools nationwide in 2009, then all public and private junior high schools in 2010 and all public and private high schools in 2012. PAPS evaluates quickness, muscular strength and endurance, flexibility, cardiovascular endurance, and BMI of all students from 1st to 12th grade once a year (Figure 1). Levels between 1 and 5 are set for each component (1 being “Excellent”, 5 being “Very Poor”), and the Ministry of Education encourages students in levels 4 and 5 to plan and engage in physical fitness programs at schools, and all schools implement their own physical fitness programs based on the PAPS results (Figure 1, see Appendix A)

The evaluation components of PAPS include a comprehensive approach to monitoring Health-related Physical Fitness (HrPF), including assessments of body composition (with an eye towards early intervention in cases of obesity) and posture, as well as a thorough psychological test [10]. Prescriptions for physical activity levels are provided after the comprehensive PAPS assessments are completed. For purposes of the PAPS evaluation, a student’s HrPF level is ranked on a scale of 1 through 5 for each assessment area, with the most physically fit students placed in the 1st level, and less fit students falling within levels 4 and 5. Less fit students are encouraged to increase their physical activity [11].

According to publicly available data from a school information disclosure site, the number of students who received PAPS scores in levels 1 and 2 have decreased over the previous three years among elementary, middle and high school students, while the number of students who fall within levels 4 and 5 have increased in Seoul [12]. Among middle school students, those who fall within levels 1 and 2 made up 5.6% and 35.7%, respectively, of the student population in 2018, a decrease of 0.1% and 1.0% from the previous year. Meanwhile, students with scores between levels 4 and 5 made up 13.7% and 1.7%, respectively, of the middle school population in 2018, an increase of 1.0% and 0.2% from the prior year in Seoul [12]. The overall decrease in physical activities may be a result of a lack of individual participation in physical fitness or exercise activities, or a lack of appropriate exercise programs in physical education classes at their schools. It is recommended for children or adolescents between 5 and 17 years of age to engage in moderate-to-vigorous physical activity (MVPA) for about 60 min per day [13]. Higher recreational screen time is associated with unfavorable body composition, higher cardiovascular risk, lower physical fitness, and lower self-esteem in adolescents [14]. In Germany, 60.4% to 72.3% of boys and 55.6% to 57.8% of girls spent two hours per day watching TV and using electronic devices [15]. In Seoul, 22.4% of their 570 junior high school students were addicted to smartphones or screen devices [16]. In Korea, there has been a rapid increase in smartphone addicted adolescents, from 11.4% in 2011, to 18.4% in 2012, 25.5% in 2013, and 29.2% in 2014 [17].

Every provincial office of education, including the Seoul Metropolitan Office of Education, endeavors to improve the health of students with PAPS scores between the 4 and 5 ranges. A number of studies have demonstrated the effectiveness of health-related physical fitness classes, in both addressing the problem once it arises, and preventing the problem from emerging [18,19,20,21,22,23,24]. Kim (2002) demonstrated that students with poor HrPF scores voluntarily and actively participated in the sports they were interested in, and showed a desire to continue to participate in the exercise programs [25]. This suggests that positive health outcomes are more likely to be achieved if students with poor health are provided with exercises tailored to their individual characteristics and interests.

Various aspects of PAPS have been thoroughly studied, but until now, there has been no assessment of the effectiveness of the school based HrPF program currently recommended by the Korean Ministry of Education for students with PAPS scores of 4 and 5. In the present study, we evaluated whether this program was effective for high-risk youths. Specifically, the purpose of the present study was to evaluate if a 16-week school-based HrPF program would improve PAPS scores of middle school students.

## 2. Method

### 2.1. Study Design and Participants

The present study was a quasi-experiment design (One-Group Pretest–Posttest Design). The present study was performed in a middle school located in Seoul. School teachers or administrators encouraged all students to participate in a schoolbased HrPF program. A total of 129 students started the program. Among those 129, 42 students were classified at level 4 or 5 (for details on levels, see Section 2.2. PAPS measures) in the regular April 2019 PAPS evaluation. Among all the participants, only 36 students’ scores were analyzed for the present study; six students out of the 42 withdrew from the study because they wanted to discontinue.

The purpose and requirements of the study were sufficiently explained to all participants, and their parents or family correspondence. They were all provided with written informed consent before the study began. The Health-related Physical Fitness (HrPF) program was operated with the consent of the students and their legal guardians. The study was approved by the IRB (JNUE-IRB-2020-012), confirming the extraction of demographical and clinical data for analysis and written or verbal consent from the study participants.

Appropriate total sample size was calculated using G*power 3.1.9.7.; test family = *t* tests, statistical test = means: difference between two dependent means (matched pairs), type of power analysis = a priori: compute required sample size given α, power, and effect size, input parameters (one, effect size (0.5, medium), α err prob (0.05), power (0.8)). The calculated total sample size (n) equals 27, with an actual power of 0.81.

### 2.2. PAPS Measures

In the present study, only “Required” components of PAPS were measured, and “Elective” metrics were excluded from measurement (Figure 1); to evaluate the “Elective”, the school must own expensive equipment, but most schools do not have this equipment. An essential PAPS assessment measures cardiopulmonary endurance, muscular strength, and endurance, quickness, flexibility, and body composition. Typically, the cardiovascular endurance evaluation consists of a Progressive Aerobic Cardiovascular Endurance Run (PACER), distance run, and step test. The muscular strength and endurance evaluation normally consist of sit-ups and maximum grip strength tests, as well as push-ups. Quickness is generally measured with a 50 m-running test and a standing long-jump test. Flexibility is normally evaluated with comprehensive flexibility tests and trunk flexion tests. Body composition is commonly assessed using the body fat percent and body mass index (BMI).

In the present study, with consideration of the facilities and convenience of the school, only one test measuring each of the essential PAPS components was employed: PACER for cardiovascular endurance, trunk flexion test for flexibility, sit-up test for muscular strength and endurance, 50 m-run for quickness, and BMI for body composition. Each PAPS evaluation was taken twice, once before and once after the HrPF program was implemented.

In the PACER test, one lap is a 15 m distance. The recorded score is the total number of laps completed by a student. The student ran as many laps as possible. In the trunk flexion test, sit-and-reach was evaluated using a sit-and-reach testing equipment (K-115, KL Sports Industry Co., An Yang, Korea, http://www.klsports.co.kr/ (accessed on 22 March 2021)).

In the sit-up test, the student performed the test with bent knees, feet flat about 45 cm from the buttocks using a testing equipment (K-111, Sports Industry Co., An Yang, Korea). The student touched their elbow to the sensor with each sit-up. The students performed as many sit-ups in 1 min as possible.

In the 50 m-run test, the student stood at a starting line and waited for a start signal. Once the student started running, the student ran as fast as possible for 50 m straight.

For the BMI test, each student’s body weight and height were measured.

### 2.3. Health Related Physical Fitness(HrPF) Program

The 36 participants were enrolled in the HrPF recommended (as of 2019) by the Seoul Metropolitan Office of Education for 16 weeks, during which they met for three 50-min sessions per week.

Each 50-min class was comprised of a warm-up exercise (5 min), exercises related to four physical fitness elements (10 min each), and a cool-down exercise (5 min) (Table 1). Because students with poor physical fitness were enrolled, exercise intensity was initially low and proceeded according to their personal fitness level but was gradually increased over the course of the 16-week program. In addition, efforts were made to maintain a positive and energetic class atmosphere to sustain the participants’ interest in the exercise program. The exercise levels were determined by the rating of perceived exertion (RPE) using a scale from 0 to 10, where sitting is 0 and the highest level of effort is 10. The exercise intensity level was gradually increased on an RPE scale of 3~4 (Light) to an RPE scale of 5~6 (Moderate) [3].

### 2.4. Statistical Analysis

Demographic characteristics of the study subjects were listed in Table 2. In order to analyze the effect of the HrPF program on PAPS measures, a paired sample *t*-test was used (SPSS 22.0). The statistical significance level was set to α = 0.05. Descriptive statistics, effect size, and confidence interval (CI) for the difference between means were computed for additional analysis (Table 3).

## 3. Results

Demographic characteristics of the study subjects are shown in Table 2.

Descriptive statistics (mean and standard deviation) and overall scores of the variables used in this study were analyzed, and the results are shown in Table 3.

### 3.1. Cardiovascular Evaluation

There was a significant increase in PACER in the post-test (95% CI; 16.579, 20.701) compared to the pre-test (95% CI; 11.736, 14.984), t(35) = −9.88, *p* < 0.001 (Table 3). The 95% confidence interval for the difference between the means was statistically significant (95% CI; 2.608, 7.952).

### 3.2. Muscular Strength and Endurance Evaluation

There was a significant increase in the number of sit-ups in the post-test (95% CI; 22.434, 27.346) compared to the pre-test (95% CI; 15.702, 20.418), t(35) = −14.26, *p* < 0.001 (Table 3). The 95% confidence interval for the difference between the means was statistically significant (95% CI; 3.365, 10.295).

### 3.3. Quickness Evaluation

The results indicated that there was not a statistical difference between the pre-HrPF 50 m-Run (95% CI; 10.315, 11.165) and post-HrPF 50 m-Run (95% CI; 10.282, 11.098) results, although a small improvement was apparent in program participants (Table 3). The 95% confidence interval for the difference between the means was statistically insignificant (95% CI; −0.549, 0.649).

### 3.4. Flexibility Evaluation

The results indicated that there was a statistical difference between the pre-HrPF Trunk Flexion (95% CI; 1.500, 3.780) and post-HrPF Trunk Flexion results (95% CI; 5.072, 6.888), t(35) = −14.00, *p* < 0.001 (Table 3). The 95% confidence interval for the difference between the means was statistically significant (95% CI; 1.857, 4.823).

### 3.5. BMI Evaluation

The results indicated that there was not a statistical difference between the pre-HrPF BMI (95% CI; 23.543, 26.137) and post-HrPF BMI (95% CI; 23.581, 25.939) (Table 3). The 95% confidence interval for the difference between the means was statistically insignificant (95% CI; −1.704, 1.864).

## 4. Discussion

As a general matter, physical fitness has a skill-related and a health-related component. In recent years, the health-related component has received more attention [15,16,17,26]. Cardiovascular endurance, muscle strength and endurance, quickness, body composition, and flexibility are all elements of health-related physical fitness that are deeply tied to daily exercise habits [27,28,29]. Particularly in adolescents, physical fitness should emphasize health-related concerns, because these aspects of fitness can be seriously degraded by a lack of exercise, significantly increasing the risk of health-related disorders or diseases [27,30,31,32]. In South Korea, adolescents perform less and less physical activity as a result of their educational environment, which is focused on examinations, and increasing time spent on smart devices, despite the fact that the middle school years are among the most critical for physical development and growth [12,33].

In agreement with previous studies [34,35,36], the study participants in the present study improved their cardiorespiratory endurance. Previous studies suggested that various types of aerobic exercises improved the cardiovascular endurance of adolescents and as cardiorespiratory endurance decreased in adolescents, the incidence of obesity increased [37]. To improve their cardiorespiratory endurance, individual students would have to maintain regular exercise habits. To achieve this, schools would have to introduce programs specifically geared towards improved cardiorespiratory endurance.

Similar to previous studies [38,39,40,41], study participants improved their muscular strength and endurance. In addition, Smith et al. [42] reported that adolescents who exercised less, watched more television, and had a higher body mass index had poorer muscular strength and endurance [42].

In a deviation from previous studies [43,44,45], participants in our study did not improve their quickness after the 16-week program. This may be because previous studies mostly tested professional athletes, and not students with poor PAPS scores. Quickness as a characteristic is difficult to significantly improve through training, as it is, to a large degree, influenced by genetic factors and not simply physical strength. As the HrPF program of this study was undertaken with the goal of improving overall physical strength in participants, it may not be reasonable to expect rapid improvement in quickness over a mere 16 weeks.

In agreement with previous research [46,47,48,49,50], the participants’ flexibility improved. Prior studies found that adolescents who regularly exercised demonstrated good flexibility. Alaguraja and Yoga [46] suggested that good flexibility allowed participants to achieve efficient motor function, stabilize and smoothen their movements, improve exercise performance, and reduce the risk of future injuries [46,47,49].

The participants in the present study did not improve their BMI. Similar results were found in the previous studies [51,52,53] that reported no changes in weight, body fat mass, and lean mass after participation in various exercise programs. Of course, the goal of the fitness program in our study was not to achieve weight loss, but rather to improve overall physical fitness. Participating students received no coaching regarding their diet for the purpose of weight loss. In fact, we did not measure body fat mass, but rather body mass index; had we measured body fat mass (%fat), the results may have shown an improvement.

PAPS evaluates body composition through the % body fat or BMI. One study [54] that investigated the relationship between BMI and physical ability, noted that adolescent obesity inversely correlated with overall motor development, while an active lifestyle was associated with an improvement in motor development [55,56,57]. These results were not confirmed in the present study. This may be a sign that BMI should be used to measure weight status in population or to screen potential weight problems in individuals.

Adolescents should participate in at least 60 min of physical activity daily to decrease the risk for numerous adverse health outcomes [3], and activity programs provided by schools should be a key strategy to increase physical activity levels and address health issues for adolescents [58]. In spite of these recommendations and suggestions [3,37,38,39,40,41,46,47,48,49,50,54,55,56,57,58], middle school students’ regular physical education classes are only 2~4 credit hours per week in Korea, England, Japan, Finland, Canada, New Zealand, France, among other countries [59,60]. These hours for physical education classes for middle school students are very short to meet physical activity needs to maintain good health outcomes [57,58,59,60]. School-based interventions by teachers or staff can help provide students with consistent messaging and opportunities related to physical activity; as a result, they are very effective in helping to foster positive attitudes toward physical activity [57,58,59,60,61]. A study [62] reported multiple systematic reviews of school-based programs on the outcomes of physical activity levels and BMI or obesity prevention. The study [62] suggests that school-based physical activity interventions are associated with some positive effects and should be focused on fostering positive attitudes toward physical activity and geared toward promoting physical activity within school-based interventions. Few information about the effects of school-based HrPF programs in Korea on health outcomes of adolescent and physical activity measures is available internationally, although data from Belgium, Canada, Czech Republic, Cyprus, Estonia, Germany, Hungary, Italy, Spain, Sweden, and the U.S.A are available [60,61,62,63,64]. The results from the present study suggest that one hour of HrPF class in addition to regular physical education classes in a day in a school setting should improve HrPF levels or PAPS measures of middle school students in Korea.

The present study included all the PAPS evaluation items during a 1-h physical fitness program, whereas most of the school-based physical activities from the previous studies were an extension of already existing regular weekly physical education classes, including several short activity breaks (2–5 min each), physical activities during recess, or flyers and posters to promote active commuting to school [65,66]. However, in the results, both the present study and previous studies suggest that the interventions are effective in improving HrPF of adolescent.

There were limitations to our present study that could be addressed in future research. Due to classroom or educational situations, the factor of interest was not manipulated and no control group was assigned for the present study. In other words, the present study was a quasi-experiment. Therefore, the major limitation of the present study is that it lacks a demonstration of cause due to there being no control group. Simply put, the present study is prone to threats to internal validity such as testing effects and regression toward the mean. First, study results were probably influenced by the home environment, dietary choices, and personal lifestyles of the students, factors that were beyond our ability as researchers to control. However, any factors related to PAPS measures were attempted to be controlled as much as possible in the present study. For example, participants were told not to change eating habits or/and daily schedule. The results from the present study suggest that there were no effects in physiological growth and development over the 16 weeks; participants’ growth in terms of mass and height were not significantly changed after 16 weeks. Second, the results were derived by subjecting all participants to the HrPF, and assessing their progress with pre- and post-HrPF tests. However, future studies would benefit from dividing the participants into experimental and control groups. Third, we did not attempt to restrict participants’ access to other physical activities; thus, in addition to the HrPF program, the observed changes in the PAPS factors may have been driven by participation in physical education classes or other school-offered physical activities. Nevertheless, there was very little chance that subjects in the present study would participate in any other after school exercise program besides the program offered by the present study, since class was over at 4:30 p.m. and the program lasted until almost 6:00 p.m.

## 5. Conclusions

Individual health care plans should focus on identifying and managing basic health-related lifestyle habits, including disease prevention, eating, exercise, rest, and sleep. Ideally, of course, middle school students would be self-aware of their health choices, but schools have a significant role to play, and should encourage students to regularly participate in health-related classes and physical activities. In particular, participation in a moderate level exercise should be effective in improving overall fitness, promoting the development of physical and motor skills, and enhancing the body’s ability to cope with diseases and risks. Because of the emphasis placed on academic achievement and the frequent use of smart devices among Korean middle school students, this group maintains a low level of physical activity. To maintain and promote good health among these students, they should be engaging in physical activities that increase their heart rate for at least 30 min a day, three days a week. If this proves to be insufficient, middle school students should explore and practice alternative means of increasing their health-related physical activity throughout the school year. Specifically, pursuant to current recommendations by the Office of Education, a systematic plan for health and fitness classes geared towards students with poor physical fitness should be established and then implemented at the school site.

## Figures and Tables

**Figure 1 healthcare-09-00358-f001:**
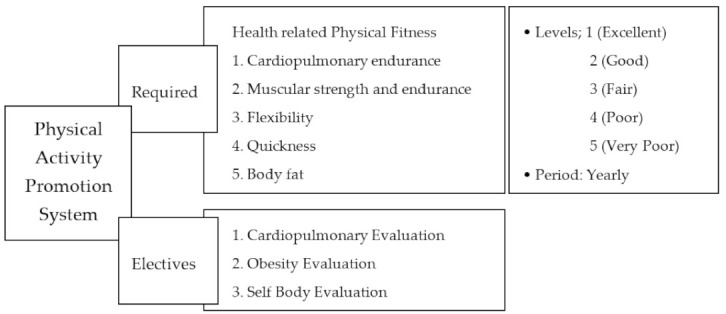
Schematic Diagram of Physical Activity Promotion System (PAPS).

**Table 1 healthcare-09-00358-t001:** Contents of HrPF Program.

Steps	Exercise Elements	Durations
Warm-up	Stretching Exercises, Walking and Jumping	5 min
Fitness exercises		
Cardiovascular endurance	Jump Rope, Cycling, Interval Training	20 min
Flexibility	Trunk, Hip, and Leg Flexions and Extensions	
MusStrEnd	Push-ups, Sit-up, Squat, Burpee	20 min
Quickness	30 m run, Sergeant Jump, Consecutive Obstacles Jump	
Cool-down	Walking, Stretching Exercises	5 min

MusStrEnd: Muscular Strength and Endurance.

**Table 2 healthcare-09-00358-t002:** Demographic of the study sample (*n* = 36).

Variables	Participants (*n* = 36)	Percentage (%)
Sex		
Male	18	50.0
Female	18	50.0
Grades		
7th	11	30.6
8th	11	30.6
9th	14	38.8
PAPS level		
4	28	77.7
5	8	22.3

**Table 3 healthcare-09-00358-t003:** Descriptive Statistics for Five PAPS Measures (*n* = 36).

Measures	Time	Mean	SD	Cohen’s D	*p*-Value
PACER (repetitions)	Before	13.36	4.98	0.93	<0.001
	After	18.64	6.31		
Sit-up (repetitions)	Before	18.06	7.22	0.93	<0.001
	After	24.89	7.52		
50 m-Run (seconds)	Before	10.74	1.30	−0.04	0.063
	After	10.69	1.25		
Trunk Flexion (centimeters)	Before	2.64	3.49	1.06	<0.001
	After	5.98	2.78		
BMI (kg/m^2^)	Before	24.84	3.97	−0.02	0.458
	After	24.76	3.61		

SD: standard deviation, PACER: Progressive Aerobic Cardiovascular Endurance Run, BMI: Body Mass Index.

## Data Availability

Restrictions apply to the availability of these data. Data was obtained from Seoul Metropolitan Office of Education and are available from the authors with the permission of Seoul Metropolitan Office of Education.

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
