# Peer review of "Effects of School-Based Exercise Program on Obesity and Physical Fitness of Urban Youth: A Quasi-Experiment"

_healthcare, 2021, doi:10.3390/healthcare9030358_

Round 1

Reviewer 1 Report

After a new review of this work, I find it appropriate to be published, I think the changes have turned out to be enriching and giving more clarity to the article, congratulations. 

Author Response

Thank you.

Reviewer 2 Report

This is a nice paper on the association of school exercise with obesity
and physical fitness. Strengths of the paper are the different
assessments conducted. The paper is well written and the results of
interest. Please find below my remarks that may be helpful in further
improving the manuscript.

1. Was there a control group? If yes, what type of control? If no
control group, the conclusion that fitness improved as a result of the
fitness program is not valid. Changes may have occurred just because of
time and because a low-performing group was examined which just show
improvements as a result of regression to the mean.

2. The sample is very small. Did the authors do a power analysis to
justify? Are estimations reliable in such a small sample?

3. The study seems to include only high-risk youth showing poor initial
health status. As said, improvements are expected just by chance. Do
patterns hold for other fitness levels at baseline?

4. Is the size of mean-level differences meaningful in real life
performance/health?

5. Were patterns comparable between the different age groups?

6. Were the patterns similar in girls and boys?

7. The authors may want to discuss in more detail the applied
implications of the findings.

8. The results need to be put in context. What does the study add to the
bulk of already existing research on fitness programs?

9. What about other societies/geographic regions? Do authors expect
differences? Or do results only hold for urban areas?

Round 2

Reviewer 2 Report

Thanks for the revision. 

I don't have further questions about this paper. 

Author Response

Thank you for your review.